# Development and Validation of an E-Learning Education Model in the COVID-19 Pandemic: A Case Study in Secondary Education

Mónica Martínez-Gómez [1], Eliseo Bustamante [1,*] and César Berna-Escriche [1,2]

1   Departamento de Estadística, Investigación Operativa Aplicadas y Calidad,
    Universitat Politècnica de València, 46022 Valencia, Spain
2   Instituto de Ingeniería Energética, Universitat Politècnica de València, 46022 Valencia, Spain
*   Correspondence: elbusgar@eio.upv.es

**Abstract:** E-learning was crucial during the global lockdown. In this way, this article aims to propose and validate a holistic framework in which all the E-learning services are needed to ensure their effective implementation and use. To this end, an original 3S-T model, to measure E-learning success based on self-student assessment, was developed. This innovative model, which reinforces the existing theoretical framework of models, identifies a wide array of success predictors and relates them to various measures that help to reach success, including learning and academic achievements. The validation of the 3S-T model was carried out using the partial least squares structural regression equations modeling technique (PLS-SEM). In this analysis, four major constructors were identified as determinants of E-learning service performance, namely, the surrounding conditions, system characteristics, tutor's development and student's own performance. Although each of them is composed of several subcategories, finally, 15 indicators that estimate the fulfillment of these factors were defined and evaluated. The present study is strongly connected to the fourth goal of the Agenda established by the United Nations, which seeks Quality Education to ensure the sustainable development of countries.

**Keywords:** E-learning; methodology; statistical analysis; student self-assessment; student satisfaction; partial least squares (PLS); COVID-19

## 1. Introduction

The unstoppable advance of new information and communication technologies (ICTs) has led to important changes in many disciplines [1,2]. These technologies have taken advantage of new paradigms such as the internet, social networks, cloud computing, block-chain, and big data to favor their unstoppable expansion. These innovations have led to the emergence of new markets, products, processes, and services. The education field has been one of the disciplines where these technologies have had an important effect; many new paradigms were brought into learning, such as E-learning and mobile learning, meaning that the existing traditional face-to-face master classes are no longer the only learning option. In fact, the E-learning paradigm is not new, but it is considered an extension of the distance education mode initiated in the 1980s [3]. This process towards the increase in distance education has been accelerated by COVID-19, since E-learning has proven to be the only resource that allows learning to continue during the nowadays global blockade [4,5]. As a result, all institutions worldwide have been investing extensively in E-learning, so that most of the courses provided in traditional face-to-face mode have been converted to E-learning mode. This acceleration has only shortened a process that was already inevitable, so that in the near future, with the return to normality, the road traveled will not be undone and nothing will return to the way it was in the past; thus, E-learning is here to stay. Additionally, according to sustainability criteria, E-learning compared to

conventional face-to-face learning has a much lower average energy consumption and $CO_2$ emissions per student [6].

In the above-described scenario, to favor this E-learning, an online learning environment (OLE) has to be created; this means that Learning Management Systems (LMSs) have to be implemented. Within them, a key factor to achieving success is the deployment of a high Quality of Interaction (QoI) between the different actors of the online learning system. A widely used LMS tool is the Moodle platform [7]; in fact, this is the tool used by the student of the current research. All these platforms must offer fast access and a user-friendly environment, with large data management capacity and a variety of web-based tools. In the case of Valencia Community (Spain), Aules [8] is the official website for E-learning, developed by and for teachers of the correspondent institutional Department of Education (Conselleria d'Educació, Generalitat Valenciana). Any teacher of the public non-university educational system of this region can create and manage a virtual class with their students. Aules is based on a Moodle platform; it is very intuitive and attractive, taking into account that the age ranges of the students are from primary school (4 years old) to the end course before university (2nd baccalaureate—18 years old).

E-learning systems should be seen as a breakthrough, as they can even compensate for the weaknesses of traditional learning methods, as well as offer the possibility to extend knowledge to a more significant number of students, who may even be on the other side of the world. Therefore, if advantage is taken of all the possibilities of the new technologies, it will offer an excellent opportunity for young people and knowledge finders. However, in order to know if the implementation of E-learning is indeed an improvement compared to traditional in-person learning, it is essential to use success-measurement tools, which are fundamental in understanding the added value, the effect of management operations on the investments [9,10]. Student Self-Assessment (SSA) is a valuable way to evaluate E-learning success; this term could be understood as a person's perceived quality of his work and educational abilities [11]. When aiming to evaluate the SSA, multidimensional factors must be analyzed in order to assess the different aspects involved in their learning [12].

The measurement/estimation of the effectiveness of E-learning initiatives has been vastly investigated [2,3,13]. A quick review of the publications in this field reveals that different studies use different conceptual approaches, such as the Technology Acceptance Model (TAM) [14,15], Information Systems Success (ISS) [9], SERVQUAL [16], the Decomposed Theory of Planned Behavior (DTPB) [17], the UTAUT and UTAUT2 models [18,19], and the 5Q model [20]. In addition, many E-learning success and quality evaluation models have been proposed, such as the E-Learning System Success (ELSS), the Evaluating E-learning Systems Success evaluation (EESS), the E-Learning Quality (ELQ), the E-learner Satisfaction (ELS), and the User Satisfaction Model (USM) [21–27]. In the same way, along with these researchers, many different dimensions, factors, and constructs have been considered to evaluate the E-learning performance in each particular application. Thus, several factors have been found to be critical to E-learning success. This paper presents a robust model for the measurement of E-learning systems performance based on students' academic achievements and on students learning attainment. We are aware that it has also been difficult for educators but there are many papers that reveal a positive relationship between teacher's perception and technology acceptance (E-learning) [28–31]. To this end, the fundamental purpose of this study is the development and validation of a tool that provides data to help better understand the factors that influence E-learning of students, and not only this, but also to be able to estimate the importance of each of them. In this model, not only was the measurement model for each construct validated, but the relationships between the measurement model and the structural model were also determined.

## 2. General Overview of the Non-University Educational System of Spain

It is crucial to understand the structure and situation of the non-university educational system for the public and potential readers of the article because the general knowledge is very low due to two fundamental aspects:

Each political option tries to impose new legislation repealing the previous one with sudden significant changes. In this vein, each government enacts a new educational law. Besides, all the communities of Spain have the possibility to modify from 40% to 50% of the contents (depending on if the community has its own language or not).

From the university and the business world, the structure and curriculum of non-university education is generally unknown. It seems rather isolated when there should be strong relationships and mutual knowledge. Obviously, non-university students go to the world of work or to a university.

The absence of a consensus among political parties to create a solid general law that would be valid for all political options and that would remain in force for a long time is highly criticized. In addition, the laws are increasingly lax in terms of the levels of demand for students. Furthermore, according to the above mentioned, local governments can be different from the central Spanish government and the assigned percentage can vary in one side or another, resulting in very different curricula. In Spain, the next course will begin to apply the new law LOMLOE [32], promoting the universal design of learning, the use of ICTs, and multilingualism among other educational aspects. The students during the COVID 19 pandemic and to date have been subjected under LOMCE [33], which has recently been repealed. Both laws divided the students into two large groups: compulsory (ESO: 1st, 2nd, 3rd, and 4th); and non-compulsory (high school or baccalaureate: 1st and 2nd). In the world, the age of leaving the non-university system is around 18 years, although there are exceptions, such as Italy, where they remain one more year in high school. Spain is one of the countries with the shortest baccalaureate (2 courses). Until the 1990s, with the General Law of Education (LGE, BOE 6-8-1970) [34], the Spanish baccalaureate had four years and the last course was especially oriented towards university (COU University orientation course). According to some teachers, a baccalaureate of only two years is too short to achieve the knowledge, strategies, and skills to begin university.

Currently in Spain, the non-university system is made up of the following:

- Child education (0–6 years), first cycle (0–3 years), and second cycle (4–6 years).
- Primary education. (1st, 2nd, 3rd, 4th, 5th, and 6th) (6–11 years old).
- Compulsory secondary school (1st, 2nd, 3rd, and 4th ESO) (11–16 years old).
- Non-compulsory secondary school (1st and 2nd) (16–18 years old).
- Vocational training cycles (intermediate and higher) (more than 16 years old).
- Other special regime teachings: languages, arts, dance, sports.

This article focuses on secondary students: compulsory and non-compulsory. This age (adolescents, teenagers) is very complicated and the effects of E-learning during the COVID-19 pandemic needs to be analyzed in depth. Students of this age are very sensitive, and they are the future power of a country. However, there is a considerable lack of articles focused on the secondary education period, when this period is crucial for the same secondary analyses or first-year university students.

## 3. Theoretical Framework

E-learning is the most widely used educational method for accessing remote resources with the help of computers, laptops, cloud systems, internal networks, tablets, and smartphones. The utilization of the latest technologies provides an added advantage in education, in the teaching–learning framework. E-learning has many advantages over traditional forms of learning; among others, it could be mentioned that it has greater accessibility to teaching material, provides fast and fluid communication, and facilitates the possibility of academic collaboration among students and the teacher. The continuous technological innovation and the vast advances have contributed to the difficulty in finding a sole E-learning definition. For example, E-learning could be defined as the use of technology during the learning process [22,35] or an information system that can incorporate a diversity of didactic material through e-mail, discussion, assignment, tests, and real-time online chat sessions [36,37]. Similarly, there are different methods to evaluate the success of an E-learning system; for instance, the Information Systems Success Model (ISSM)

by [9], the Technology Acceptance Model (TAM) by [38,39], the User Satisfaction Model (USM) by [27,40], the E-Learning Quality (ELQ) models by [22,24], among many other different models.

In order to provide a global definition applicable to the different methodologies for measuring E-learning success, various theories and acceptance models have been consulted. Thus, in the elaboration of our model, mainly four of the most widely used approaches to evaluate E-learning and information systems have been considered, in such a way that it brings together the different contributions of each one of them.

### 3.1. Approach 1: Information System Success Model (ISSM/D&M Model)

The ISSM model was first introduced by [9] in the 1990s, being one of the best-known models for evaluating information systems' success. It proposes the use of six interrelated and interdependent variables to evaluate the success of non-face-to-face learning systems: system quality, information quality, usage, user satisfaction, individual impact, and organizational impact. The quality of the system will be given by the desirable characteristics of the system (reliability, speed of access, etc.). Information quality refers to the timeliness, accuracy, completeness, clarity, etc., of the information contained in the platform. Usage refers to the users' perception of use adequacy to perform the different tasks to be carried out in the system. User satisfaction can be defined as the degree of satisfaction that the user has during the use of the system. Individual performance could be defined as the individual perceived gains by the users during the use of the system, mainly in relation to their educational skills. Organizational impact focuses on users' perceived level of organizational system success.

The authors modified their original model [9], since relevant criticisms were received over several years. The enhanced model adds the variables of intention to use and service quality and replaces the individual and organizational impact variables with the net benefit variable. For the authors, quality of service refers to the support quality received by the users from the system service provider; in turn, intention to use is defined as users' predisposition to keep using the system, whereas net benefits are defined as the final degree to which information systems contribute to the successful outcomes of individuals, groups, and organizations (allowing researchers to apply the ISSM model to the desired level of analysis that is most appropriate to the research context).

A review of the existing literature on online education indicates that there is a consensus on the model's validity, at least partially, for evaluating the performance of E-learning systems. Nevertheless, there are contradictions in the results when comparing different studies. For instance, whereas some research reported a strong effect of the general quality issues (service quality and system information) on current system utilization, other authors pointed out that this relationship was negligible.

The constructs taken from this model are the following:

1. System Quality (SQ).
2. Service Quality (SEQ).
3. Information Quality (IQ).
4. Student Satisfaction (SS).
5. Student Academic Performance (SAP) (Benefits).

### 3.2. Approach 2: Technology Acceptance Model (TAM)

Davis' Technology Acceptance Model (TAM) [41] is another possible early model to evaluate the acceptance of information systems. This theory has been the most extensively used for measuring the success of a new technology in terms of its acceptance and use. This approach was defined from the Theory of Reasoned Action (TRA) and was classified within the theories of Social Psychology. The model is based on the fact that when users are presented with a new technology, several factors influence their decision on how and when they will use it. According to this model, external, social, cultural, and political factors are determinants in estimating the perceived usefulness and perceived ease of use by the user.

Additionally, user-perceived usefulness and ease of use are the main predictors of attitude towards the use of the technology and intention to use it, while the intention to use is the most important indicator of current system use. In this work, ease of use, according to [22], is not considered as a separated construct due to the relationship with technical system quality (SQ).

An important number of research works based on the TAM model, as well as some of its multiple extensions, have been conducted in recent decades. For instance, an important extension, TAM2, was introduced by [41,42]. They extended the initial model by the addition of the processes of social influence (subjective norm, voluntariness, experience, and image). Instrumental cognitive processes were also considered (relevance of work, quality of results, and demonstrability of results). Years later, [18] constructed the Unified Theory of Acceptance and Use of Technology (UTAUT), which significantly improved the explanatory power of variance in intention to use. Successive extensions of TAM have evolved over time, in particular, Venkatesh published a new model, TAM3 [43] and UTAUT2 [19]. The TAM model and its different variants have been assiduously used in the context of E-learning systems to forecast the usefulness, intention to use, and usage of E-learning systems. In [41], the authors ascertained the importance of the role of Perceived Enjoyment (PE) in predicting computer acceptance and usage, and they found that PE could influence the Intention of Use (USE).

The constructs adapted from this model are as follows:

6. Perceived Usefulness (PEU).
7. Perceived Enjoyment (PE).
8. Intention to Use (EUS).
9. Subjective Norm (SN).
10. Social Networking (NE).
11. Student Learning Achievements (SSA).

### 3.3. Approach 3: E-Learning Self-Acceptance Measure (EIAM)

In aiming to evaluate the E-learning systems' success, a widespread possibility is to use the perceptions of the users. The EIAM evaluates users' perceptions of tutor quality, perceived usefulness, and facilitation conditions with regard to the utilization of the E-learning systems [44].

The EIAM model considers 21 items, a number that has been reached based on expert and students' opinions, as well as on the different existing methods; basically, the two families of models described above. Finally, the 21 defined contributions were arranged into four categories: quality of the tutor (eight items), perceived usefulness (four items), perceived ease of use (five items), and facilitation conditions (four items). The part of TAM called "attitude towards using" is considered in the current model as a part of EIAM, related to the quality of the learner and instructor.

The construct adopted from this approach are as follows:

12. Tutor Quality (TQ).
13. Strategy (S).
14. Engagement (E).

### 3.4. Approach 4: Online Learning Self-Efficacy (OLSE)

Self-efficacy can be defined as one's own belief in carrying out a specific task. In this sense, applying the notion of E-learning, it could be understood as the self-confidence of having the capability to perform certain learning tasks using a determined E-learning system. Five dimensions are considered in the OLSE model [45,46]. These dimensions are the user's self-efficacy to complete the online course; their ability to interact socially with classmates; to manage course tools; to interact with instructors; and to interact with classmates for academic purposes. In addition, according to the OLSE model, the role of demographic variables (such as gender, the number of online courses completed, and educational status) are aspects to consider in online learning self-efficacy [45].

The construct adopted from this approach is as follows:

15. One-self Efficacy (OSE).

## 4. Development of the Conceptual Model (Research Model and Hypotheses)

In an attempt to give a global definition of E-learning success metrics, the four approaches most widely used over the last decades to evaluate E-learning have been taken into account in the elaboration of our new model.

In general, as different researchers have repeatedly reported, student satisfaction is a very reliable indicator for measuring the success of the implementation of E-learning-based initiatives (strong relationship between students' perception of their academic performance and their degree of satisfaction in E-learning environments) [37,47–50]. Subsequently, Student Learning Achievements (SSA) has been widely used as an evaluation mechanism in the educational field [51–54]. The SSA is a powerful tool to evaluate the performance of E-learning strategies in higher education, but even more important is this evaluation in primary and secondary school, given that the students' preparation is slower, since they are in an earlier period of their training. Therefore, it is even more important to use tools that allow seeing the degree of achievement of the objectives sought, as well as the determination of the causes that help or hinder the achievement of the goals sought.

### 4.1. Research Model

As discussed earlier, there are a huge number of factors affecting E-learning, with a multitude of complex interactions among them. Although, it is relevant to be aware that E-learning is an efficient mean for the teaching–learning process in the current educational environment, and even more so considering the pandemic situation. However, even more important is to know, in depth, the different factors that motivate users to accept and take full advantage of the capabilities of E-learning.

In the current study, the major factors or dimensions that were identified to be determinants in the E-learning achievements, were the ones related to social aspects, student factors, system factors, and tutor capabilities; so, we renamed our tool the 3S-T model. In this model, the student factors are divided into three sub-factors, individual factors, namely, the user's beliefs, technology acceptance, and, finally, the student's own performance—dimensions that cover the main elements of the existing approaches and are the major components of our new 3S-T model, although some of them can be subdivided into several subcategories. Ultimately, there are a total of 15 constructs that contribute to the SSA.

Figure 1 represents the survey model used in the current research, relying on the four aforementioned approaches.

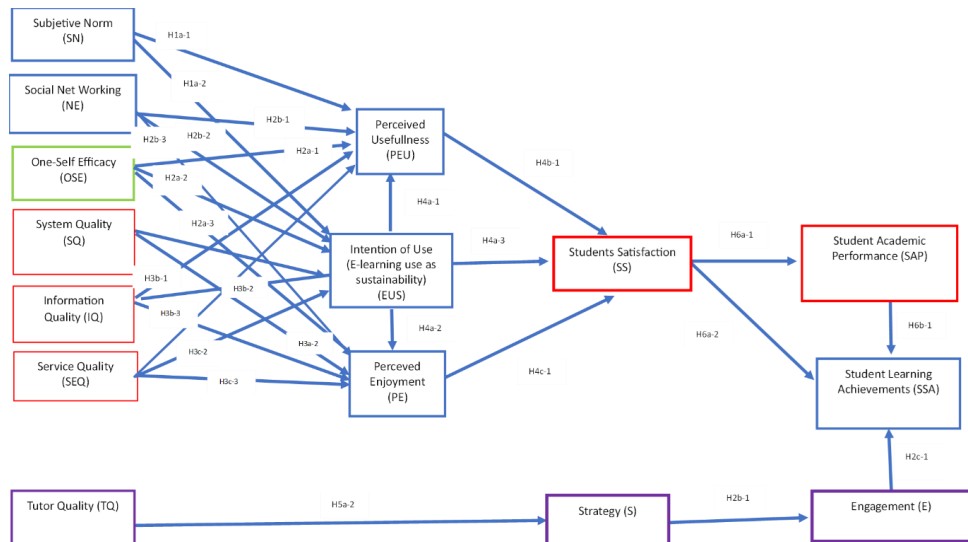

**Figure 1.** The research model (3S-T model).

### 4.2. Research Hypothesis

This section presents the linkages hypotheses in the current proposed model with the accompanying discussions. Every connection between the constructs in the model is justified on the basis on the empirically proved hypotheses found in the literature on the effectiveness of E-learning and information systems.

#### 4.2.1. Facilitating Conditions: Social Factors and Social Networking

Facilitating conditions has been defined as an external factor that accompanies the main TAM-based construct and is defined as the level up to which students perceive that organizational and technical factors exist to support the use of E-learning during the pandemic [55].

It is generally admitted that social factors (social influence) affect users' behavior very significantly [56,57]. Consequently, social factors have been tested over the years as subjective norms about the user's intention towards the faced situation [57–59]. Subjective norm is defined as "the person's perception that salient social referents think he/she should or should not perform the behaviour in question [58]". Therefore, applying this aspect to the E-learning context, when an individual perceives that his/her salient referents think that he/she should use the E-learning system, he/she will incorporate these beliefs of his/her referents into his own beliefs [57,59] and consequently will intend to use it [57]. Additionally, [60] showed that these social influences have a direct impact on the attitude and intention to adopt information technologies (IT), and further stated that users may feel compelled to participate because they may want to be part of a community. However, the enjoyment perceived by individuals usually occurs in an autonomous context; therefore, it is very likely that they are not influenced by the individual's relevant people, such as family, friends and mates [61]. Hence, it is possible that social factors may not affect perceived usefulness (PEU). Consequently, we put forward the following research hypotheses, aiming to estimate the degree of correlation, if any:

**H1a-1:** *SN will positively affect the PEU of the E-learning system.*

**H1a-2:** *SN will positively affect using an E-learning system for sustainability.*

Social Networking (NE) is another of the possible facilitating conditions; it refers to the growth of the perceived value of a product with an increase in the number of users [62,63], since the user's perceived product utility is strongly affected by the number of users using it [64]. Researchers [60] analyzed the consequences of NE on IT adoption from the point of view of the existence of a "perceived critical mass", noting that when the user perceives that this critical mass is reached by the product, it dominates the user's attitudes. In this regard, in the E-learning environment, [57] indicated that when students perceived that a large and growing number of peers were using the E-learning system, it was inevitable that they would try the system. Furthermore, [57] demonstrated that NE, on the one hand, exerted a significant direct effect on PE (perceived enjoyment), PEU (E-learning use as sustainability-intention to use), and behavioral intention. On the other hand, individuals may have an autonomous perception of their own enjoyment of the activity itself; thus, they are usually not biased by others [61]. Consequently, it is possible that SN does not affect PEU; hence, we put forward the following hypotheses to corroborate a possible correlation:

**H1b-1:** *NE will positively affect the PEU of the E-learning system.*

**H1b-2:** *NE will positively affect attitude towards using an E-learning system (EUS).*

**H1b-3:** *NE will positively affect attitude towards using an E-learning system (PE).*

#### 4.2.2. Individual Factors

Individual factor would probably be an important aspect to be considered in E-learning. Many previous research studies reported that users' own individual factors have a meaningful effect on how they perceive an E-learning system, consequently greatly

affecting their willingness to accept it. In relation to individual factors, self-efficacy reflects a person's own beliefs about his or her ability to perform certain tasks successfully [65]. Similarly, several previous studies have shown that in an E-learning context, self-efficacy influences the PEU [54,57,66–68], while other researchers have shown that it can be a very important factor affecting PEU [54,67]. Therefore, this influence is hypothesized in this study, aiming to check this relationship:

**H2a-1:** *One-self Efficacy (OSE) will positively affect the EUS of the E-learning system.*

**H2a-2:** *One-self Efficacy (OSE) will positively affect the PE of the E-learning system.*

**H2a-3:** *One-self Efficacy (OSE) will positively affect the PEU of the E-learning system.*

On the other hand, many studies indicate that individual factors have a significant influence on how users perceive the E-learning system, and, subsequently, their willingness to embrace it. However, given that today's students are different from the students of the past, they want to create, use the tools of their time, share control, and make decisions. They also want to share their opinions not only in class but globally, and additionally, they seek an education that is relevant and connected to the reality around them. All this makes them more predisposed to adapt to online learning. Consequently, the conclusions of previous research studies are even more applicable in today's E-learning environment.

In this way, all constructs related to the student's interests, motivation, perceptions, etc., are key aspects to reach adequate E-learning effectiveness [2,67]. In particular, the strategy and engagement of the students play key roles in the student's performance, usually when there are high perceived enjoyment and usefulness by the students [10,69], resulting in positive user performance. Therefore, this influence is hypothesized in this study, aiming to check this relationship:

**H2b-1:** *Strategy (S) positively influences engagement (E).*

**H2c-1:** *Engagement (E) positively influences students' learning achievements (SSA).*

### 4.2.3. System Factors

Since [41] proposed the TAM, it has been postulated that system factors strongly affect users' beliefs. Subsequent studies have proved the importance of the role of system factors in predicting users' beliefs and acceptance in the E-learning context [57,68,70,71]. The characteristics of the platform used determine the information available to the student, the way in which he/she can access it, the possibilities of sharing it, the possibilities of contacting peers or teachers, the possibilities of collaborating with peers, etc.; therefore, it will be of vital importance to achieve the E-learning objectives [72,73]. Specifically, faculty, as well as peers, are extremely important resources for students to learn. However, due to the usual complexity of E-learning platforms, as well as the barriers created by non-face-to-face teaching, it becomes more difficult for students to socialize with their faculty and peers/friends and requires the use of different approaches to establish these relationships. While, it is argued that students with prior experience in online socialization are able to approach their peers and faculty more effectively on the platform due to their familiarity with the norms and approaches [2]. Therefore, in particular, a key aspect is estimating the capability to improve the teaching–learning process. Consequently, indicators of system performance are of vital importance in evaluating E-learning achievements.

In [68], the authors analyzed the actual user usage of the E-learning system for a distance educational system using the TAM model; they concluded that system factors could affect the PEU very positively. In turn, [71] classified the system factors affecting user acceptance of E-learning into system, information, and service qualities. Consequently, the following hypotheses are put forward in this study:

**H3a-1:** *SQ will positively affect the EUS of the E-learning system.*

**H3a-2:** *SQ will positively affect the PE of the E-learning system.*

**H3b-1:** *SEQ will positively affect the EUS of the E-learning system.*

**H3b-2:** *SEQ will positively affect the PE of the E-learning system.*

**H3b-3:** *SEQ will positively affect the PEU of the E-learning system.*

**H3c-1:** *IQ will positively affect the EUS of the E-learning system.*

**H3c-2:** *IQ will positively affect the PE of the E-learning system.*

**H3c-3:** *IQ will positively affect the PEU of the E-learning system.*

4.2.4. User Beliefs and Technology Acceptance

Taking the extended TAM model [73–77] as a starting point, the proposed relationships between users' beliefs regarding the E-learning system and their subsequent acceptance and use of the system are explained hereunder. It is generally admitted that EUS directly affects the learner's attitude toward the use of the E-learning system [78–81]; in the same vein, it is admitted that PEU directly affects the attitude toward a user's use of the E-learning system [78,80,81]. Likewise, PE is directly affected by the attitude toward the use of the E-learning system [79]. In addition, EUS is considered to mediate the influence of PEU on the user's attitude toward E-learning system use [78,81]; it is also generally accepted that PE mediates the influence of PEU on the attitude toward system use [79]. The PEU directly determines the intention to use the E-learning system [54,67,82–86]; PE directly estimates the intention to use the E-learning system [78,84–86]; and attitude toward using the E-learning system directly estimates the intention to use the system [78–80]. Thus, the intention to use the E-learning system directly impacts the SS of the system [78]. Therefore, summarizing, the following hypotheses are put forward:

**H4a-1:** *EUS will positively affect the PEU of the E-learning system.*

**H4a-2:** *EUS will positively affect the PE.*

**H4b-1:** *PEU will positively affect student satisfaction (SS).*

**H4c-1:** *PE will positively affect student satisfaction (SS).*

4.2.5. Tutor's Development

In relation to the contribution of the tutor in the success of E-learning, previous research has shown that the delivery of the course, the attributes of the tutor, and the facilitator conditions of this proved to be very important, if not the main determinants, in the usefulness perceived by the students [87,88]. The role of tutors in E-learning is even more important than in traditional education, as the e-instructor must be more skilled, especially in the application of classroom technology [89]. The authors of [90] reported that with the implementation of E-learning, the role of instructors have shifted from being subject matter experts to facilitators. To succeed in an online education system, the positive attitude of tutors is crucial. In [91], the authors identified and classified the competencies possessed by e-instructors into these categories: knowledge of the online system, technical competence, communication skills, content mastery, and personal characteristics. Particularly important has been the change brought about by the closing of universities and schools caused by the COVID-19 pandemic; this change brought about various psychological changes in both students and teachers [92], greatly affecting their performance. Researchers [93] analyzed the performance of the university mentoring system during the COVID-19 pandemic. The tutor–student relationship is supported by communication and collaboration; so, not losing them requires the rapid adoption of measures that favor them in the new situation, such as the use of many communication technologies. The authors' investigations concentrated on four different forms of mentoring, namely, by email, in person, through virtual tutoring (Hangout/Google Meet), and using WhatsApp. These researchers noted that synchronous and frequent daily communication are key aspects for an efficient and successful mentoring system where the use of WhatsApp, complemented by synchronous

communication through messages and video calls, is the best form to achieve student satisfaction. Thus, we put forward the following hypotheses:

**H5a-1:** *Tutor quality (TQ) positively influences intention of use (USE).*

**H5a-2:** *Tutor quality (TQ) positively influences strategy (S).*

4.2.6. Student's Own Performance (Student Satisfaction)

Assiduity of use affects student satisfaction and performance, ultimately leading to the achievement of learning objectives. In the same way, many other correlations have been found among the different constructs of the student's own performance and with the rest of the constructs. Satisfaction has more than demonstrated its effectiveness and reliability as an essential success measurement of both information and E-learning systems [22]. In the current model, we suppose that student satisfaction is a determinant of the benefits construct; that is, student learning achievements (SSA). Therefore, the following hypotheses were put forward:

**H6a-1:** *Student satisfaction (SS) toward the E-learning system positively influences students' academic performance (SAP).*

**H6a-2:** *Student satisfaction (SS) toward the E-learning system positively influences students' learning achievement (SSA).*

**H6b-1:** *Students' academic performance (SAP) positively influences students' learning achievement (SSA).*

**5. Research Method**

Quantitative methodologies have been used to verify the theoretical 3S-T model and its hypotheses; therefore, in order to "measure the success" of E-learning through learning achievement or student academic performance (SAP) and student learning achievements (SSA) during this pandemic, a quantitative survey was adopted in this research (see Appendix A Table A1). With regard to ethical considerations, the study was approved by the local ethics committee of UPV (protocol number P03_24032022).

*5.1. Aim and Participants*

The purpose of this study was to test and improve the 62 items of the 3S-T models. These items were analyzed using a 7-point Likert response scale, in which 1 means totally against and 7 means totally in favor. Prior to its completion, the participants were informed about the objectives of the research [60].

A total of 217 students participated in this investigation. The participants in this research are students of compulsory secondary education (Educación Secundaria Obligatoria, ESO) and baccalaureate (Bachillerato, non-compulsory), consisting of four and two grades, respectively, in the age range between 11 and 18 years. The study was carried out in the city of Valencia. In Supplementary Material, we can find the answers from the students.

*5.2. Evaluation Model of E-Learning Performance*

A confirmatory analysis (CFA) with structural equation modelling (SEM) was used to examine the factor structure of the 62-item scale. Two possible SEM approaches exist: covariance-based SEM (CB-SEM) and composite-based SEM or partial least squares SEM (PLS-SEM). This study employs the second one, the PLS-SEM approach, because of the model's 15 constructs and 62 indicators. PLS-SEM is a useful approach for forecasting behaviors in behavioral research areas. For models that are complex and comprising formative (causal) and reflective (consequent) constructs, the PLS approach offers theoretical knowledge, with its major strength residing in the modelling [94,95]. This technique was chosen for its capability to simultaneously examine a series of dependence relationships, particularly when within the model latent variables of the first and second order are under study [96].

We have illustrated in Figure 2 the two stages of the methodology. Stage 1 addresses the evaluation of the reflective and formative measurement models, with both measurement models examining the measurement theory. Stage 2 addressed the evaluation of the structural model, which that covers the structural theory, involving testing the proposed hypotheses and addressing the relationships among the latent variables [97].

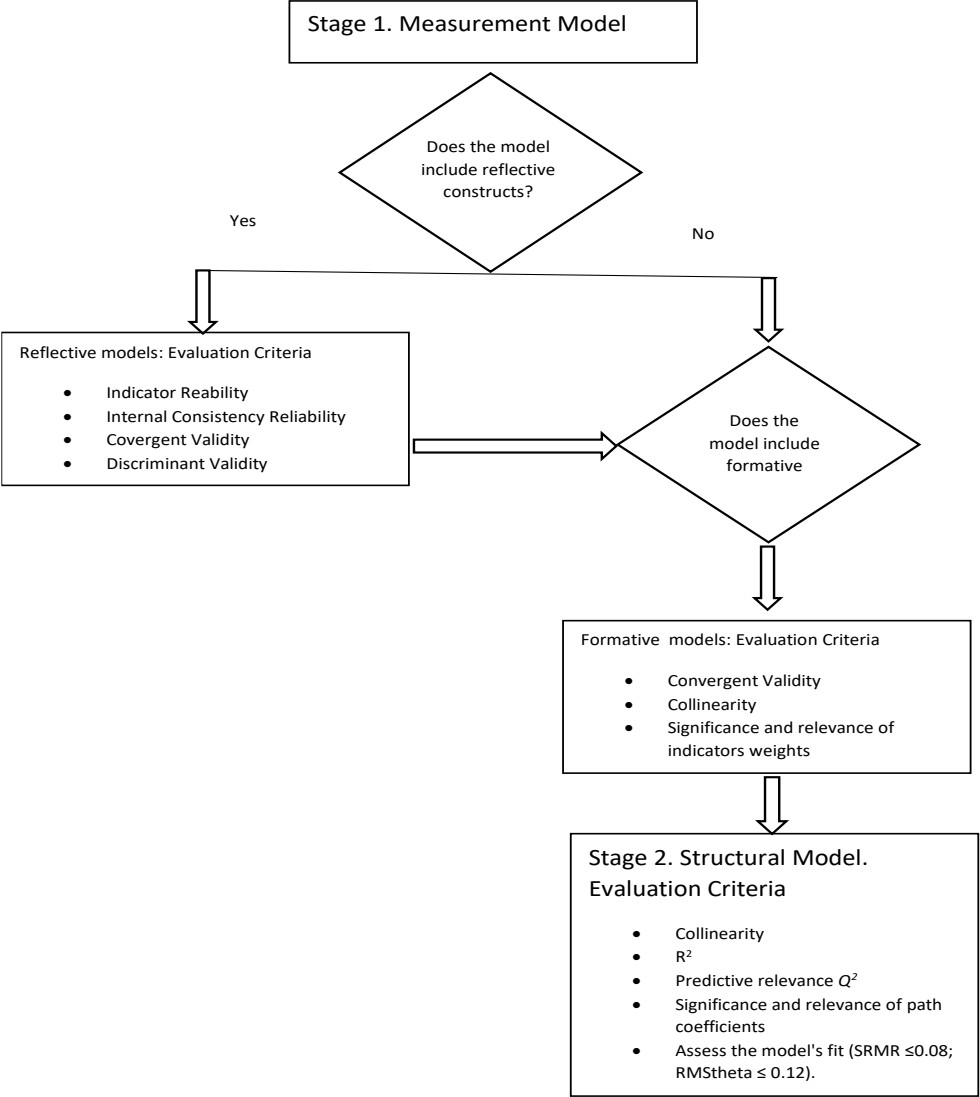

**Figure 2.** PLS-SEM model evaluation (adapted from [98]).

The model measurement and evaluation were carried out through data computation in SmartPLS 3.6. The measurement theory indicates how to measure latent variables. There are two types of measurement models [99,100]: formative and reflective measurement models.

Measurement models refer to the relationship between the indicators that reflects each construct and implies testing the measures' reliability and validity. The measurement model was evaluated through the use of the following criteria [101]:

Indicator Reliability: the outer loading for the indicator should be $\geq$0.70 [101].

Internal Consistency Reliability: using two tests, the composite reliability (CR) and Cronbach's alpha ($\alpha$). The cut off value is $\geq$0.70 for both tests [102].

- Validity:
- Convergent Validity: the average variance extracted AVE should be $\geq$0.50 [103].
- Discriminant Validity: through the use of three tests for verification:

  a. Fornell-Larcker criterion [103];

b.　　Cross-loadings [102];
c.　　The Heterotrait–Monotrait ratio (HTMT) [104].

Structural models refer to the relationship among the constructs. The structural model was assessed using the criteria proposed by [105]:

- Assess the structural model for collinearity issues (VIF < 5);
- Evaluate the significance and relevance of the structural model relationships ($p < 0.05$);
- Analyze the level of $R^2$ (the cut-off levels are 0.190, weak; 0.333, moderate; and 0.670, substantial);
- Assess the level of $Q^2$ (cut-off point larger than zero);
- Assess the model's fit (SRMR $\leq 0.08$; RMS$_{theta} \leq 0.12$).

To determine the ratio of the sample to variables for SEM analysis is also important. We can find interesting literature in this field [106–108].

## 6. Results

The average age of the students was 13.84 (SD = 1.72), and there were 94 (43.3%) females in the sample. There were 38 students (17.5%) without any computer skills. The sample characterization is shown in Table 1.

**Table 1.** Respondents' profile.

| Characteristics | | Number | Percentage (%) |
|---|---|---|---|
| Gender | Male | 123 | 56.7 |
| | Female | 94 | 43.3 |
| Age | 11–14 | 101 | 47 |
| | 14–16 | 75 | 36 |
| | 16–18 | 41 | 19 |
| Education Level | 1 ESO | 71 | 33 |
| | 2 ESO | 34 | 16 |
| | 3 ESO | 35 | 16 |
| | 4 ESO | 40 | 18 |
| | 1 Bachelor | 20 | 9 |
| | 2 Bachelor | 17 | 8 |
| Computer Skills | Yes | 145 | 68.9 |
| | No | 38 | 17.5 |
| | No answer | 34 | 15.7 |

*6.1. Measurement Model*

6.1.1. Outer Loading, Internal Consistency, and Reliability

All factors were reflective, so the outer loading was analyzed, based on the suggestion of [101], which denotes that:

- If the outer loading is less than 0.4, delete the indicator;
- If the outer loading is higher than 0.7 maintain the indicator;
- If the outer loading is between 0.4 and 0.7, analyze the impact of removing the indicator on the variance extracted (AVE) and composite reliability (CR).

In Appendix A Table A2, the outer loadings are shown. We can appreciate that some indicators fail to meet the minimum criteria, despite the indicators testing the internal consistency, the values of AVE and CR, being higher than 0.5 and 0.7 for all constructs. We analyzed the effect of removing SQ3 and USE 2, after which the values of AVE and CR declined; so, we maintained all indicators. Table 2 shows the measures of internal consistency, reliability, and validity.

**Table 2.** Measures of internal consistency, reliability, and validity.

|  | Cronbach's Alpha | rho_A | Composite Reliability | Average Variance Extracted (AVE) |
|---|---|---|---|---|
| OSE | 0.779 | 0.790 | 0.849 | 0.530 |
| PE | 0.874 | 0.874 | 0.922 | 0.799 |
| PEU | 0.888 | 0.894 | 0.930 | 0.817 |
| SEQ | 0.583 | 0.636 | 0.773 | 0.539 |
| NE | 0.783 | 0.808 | 0.872 | 0.696 |
| S | 0.791 | 0.821 | 0.856 | 0.548 |
| SSA | 0.831 | 0.832 | 0.899 | 0.748 |
| Student Academy Performance | 0.807 | 0.814 | 0.912 | 0.838 |
| Student Satisfaction | 0.787 | 0.801 | 0.875 | 0.701 |
| Subject Norm | 0.719 | 0.721 | 0.877 | 0.781 |
| SQ | 0.828 | 0.832 | 0.897 | 0.743 |
| TQ | 0.893 | 0.901 | 0.915 | 0.608 |
| EUS | 0.832 | 0.842 | 0.874 | 0.501 |
| E | 0.846 | 0.853 | 0.885 | 0.526 |
| IQ | 0.704 | 0.775 | 0.833 | 0.630 |

### 6.1.2. Discriminant Validity

Discriminant validity is the degree to which a construct is different from the rest [96]. The correlation matrix for the Fornell–Larcker method is shown in Table 3 and we can add that the diagonal values are higher than other values in the same column, which indicates that the AVE scores of every construct are lower than their is shared variance.

**Table 3.** Fornell–Larcker discriminant validity correlation matrix.

|  | E | EUS | IQ | OSE | PE | PEU | S | SAP | SN | SEQ | SS | SSA | SN | SQ | TQ |
|---|---|---|---|---|---|---|---|---|---|---|---|---|---|---|---|
| E | 0.71 | | | | | | | | | | | | | | |
| EUS | 0.70 | 0.73 | | | | | | | | | | | | | |
| IQ | 0.66 | 0.60 | 0.79 | | | | | | | | | | | | |
| OSE | 0.61 | 0.54 | 0.51 | 0.73 | | | | | | | | | | | |
| PE | 0.72 | 0.57 | 0.58 | 0.47 | 0.89 | | | | | | | | | | |
| PEU | 0.79 | 0.62 | 0.63 | 0.49 | 0.65 | 0.90 | | | | | | | | | |
| S | 0.58 | 0.46 | 0.54 | 0.51 | 0.48 | 0.51 | 0.73 | | | | | | | | |
| SAP | 0.35 | 0.33 | 0.33 | 0.34 | 0.30 | 0.24 | 0.27 | 0.83 | | | | | | | |
| SN | 0.53 | 0.60 | 0.49 | 0.49 | 0.43 | 0.45 | 0.45 | 0.22 | 0.74 | | | | | | |
| SQ | 0.72 | 0.70 | 0.57 | 0.47 | 0.60 | 0.70 | 0.46 | 0.36 | 0.47 | 0.86 | | | | | |
| SS | 0.62 | 0.60 | 0.46 | 0.44 | 0.49 | 0.62 | 0.36 | 0.28 | 0.46 | 0.68 | 0.92 | | | | |
| SSA | 0.76 | 0.62 | 0.61 | 0.48 | 0.72 | 0.74 | 0.54 | 0.34 | 0.46 | 0.69 | 0.66 | 0.84 | | | |
| SubNorm | 0.63 | 0.61 | 0.56 | 0.47 | 0.52 | 0.58 | 0.54 | 0.36 | 0.49 | 0.60 | 0.48 | 0.58 | 0.88 | | |
| SystQ | 0.58 | 0.47 | 0.55 | 0.51 | 0.44 | 0.51 | 0.58 | 0.30 | 0.40 | 0.50 | 0.38 | 0.53 | 0.48 | 0.86 | |
| TQ | 0.59 | 0.63 | 0.51 | 0.50 | 0.50 | 0.53 | 0.52 | 0.32 | 0.55 | 0.55 | 0.48 | 0.52 | 0.61 | 0.47 | 0.78 |

The second method for assessing the discriminant validity is the cross loadings (Appendix A Table A2), and we can appreciate that each indicator loads higher on the construct related to it.

Finally, we checked the Heterotrait–Monotrait ratio (HTMT) criterion proposed to [104] to assess discriminant validity, as many authors established that the criterions mentioned before are insufficiently sensitive to detect discriminant validity. Heterotrait correlations are correlations of indicators across constructs measuring different characteristics, while monotrait correlations of indicators measure the same construct. As stated, [109] the threshold values $\leq 0.9$ are accepted. The results show that the values significantly differed from 1. Table 4 shows the Heterotrait-Monotrait ratio (HTMT) correlation matrix.

**Table 4.** Heterotrait–Monotrait ratio (HTMT) correlation matrix.

|  | E | EUS | IQ | OSE | PE | PEU | S | SAP | SN | SEQ | SS | SSA | SN | SQ | TQ |
|---|---|---|---|---|---|---|---|---|---|---|---|---|---|---|---|
| **E** |  |  |  |  |  |  |  |  |  |  |  |  |  |  |  |
| **EUS** | 0.83 |  |  |  |  |  |  |  |  |  |  |  |  |  |  |
| **IQ** | 0.84 | 0.76 |  |  |  |  |  |  |  |  |  |  |  |  |  |
| **OSE** | 0.73 | 0.64 | 0.68 |  |  |  |  |  |  |  |  |  |  |  |  |
| **PE** | 0.84 | 0.66 | 0.71 | 0.55 |  |  |  |  |  |  |  |  |  |  |  |
| **PEU** | 0.91 | 0.71 | 0.77 | 0.57 | 0.73 |  |  |  |  |  |  |  |  |  |  |
| **S** | 0.78 | 0.64 | 0.85 | 0.73 | 0.64 | 0.68 |  |  |  |  |  |  |  |  |  |
| **SAP** | 0.41 | 0.39 | 0.45 | 0.44 | 0.35 | 0.29 | 0.38 |  |  |  |  |  |  |  |  |
| **SN** | 0.64 | 0.72 | 0.64 | 0.61 | 0.50 | 0.51 | 0.64 | 0.29 |  |  |  |  |  |  |  |
| **SQ** | 0.86 | 0.82 | 0.73 | 0.56 | 0.70 | 0.82 | 0.64 | 0.45 | 0.57 |  |  |  |  |  |  |
| **SS** | 0.75 | 0.72 | 0.58 | 0.53 | 0.58 | 0.72 | 0.50 | 0.36 | 0.55 | 0.82 |  |  |  |  |  |
| **SSA** | 0.92 | 0.74 | 0.79 | 0.57 | 0.87 | 0.87 | 0.74 | 0.43 | 0.55 | 0.85 | 0.82 |  |  |  |  |
| **SN** | 0.81 | 0.78 | 0.79 | 0.60 | 0.66 | 0.73 | 0.83 | 0.49 | 0.64 | 0.77 | 0.64 | 0.75 |  |  |  |
| **SystQ** | 0.70 | 0.55 | 0.73 | 0.62 | 0.52 | 0.59 | 0.86 | 0.38 | 0.50 | 0.60 | 0.46 | 0.66 | 0.63 |  |  |
| **TQ** | 0.66 | 0.71 | 0.64 | 0.59 | 0.55 | 0.56 | 0.72 | 0.38 | 0.63 | 0.62 | 0.54 | 0.59 | 0.75 | 0.53 |  |

### 6.1.3. Significance of the Outer Loading

The significance of the outer loadings was assessed through the use of the algorithm of bootstrapping in PLS. We used 50,000 bootstrap samples to estimate the t and *p* values to test the significance of the outer loadings at 5% error probability, thus meaning that the statistical significance level at 5% indicates that *p*-values must be >0.05 to accept the hypothesis and a *t*-value > 1.65. Results of bootstrapping are displayed in Appendix A Table A2. Results inform that all outer loadings are significant, with *p*-values lower 0.05.

A results summary for the measurement model assessment and the significance of outer loadings is shown in Appendix A Table A3.

### 6.2. Structural Model

As we stated previously, the assessment of the structural model includes five steps [101]. First, collinearity was assessed through the variance inflation factor (VIF). According to [105], values of VIF $\geq$ 5 indicates a potential problem of collinearity. In our case, the retrieved VIF values are all below 5; thus, our data did not present collinearity problems. Figure 3 shows the path coefficient ($\beta$ values) of the relationships between the constructs and indicators.

Table 5 shows the *p*-values obtained to assess the path coefficient between the endogenous and exogenous constructs, and by applying the same criterion of a 5% level of significance, most hypotheses were supported, while others were rejected.

The third step consists of evaluating the coefficient of determination ($R^2$) of the dependent variable, so that this measure can represent the variance proportion in the endogenous variables that can be explained by exogenous variables; i.e., it can be interpreted as the predictive accuracy of the proposed model. It ranges from 0 to 1, and [101] stated that values of 0.75 is substantial, 0.5 moderate, and 0.25 weak. As shown in Figure 3, PEU, USE, and PE performed moderately regarding student satisfaction, student academic performance (SAP), and student satisfaction and engagement (E); and moderately regarding the substantially explained (52.7%) student learning achievements (SSA). These $R^2$ results show a sufficient level of this measure.

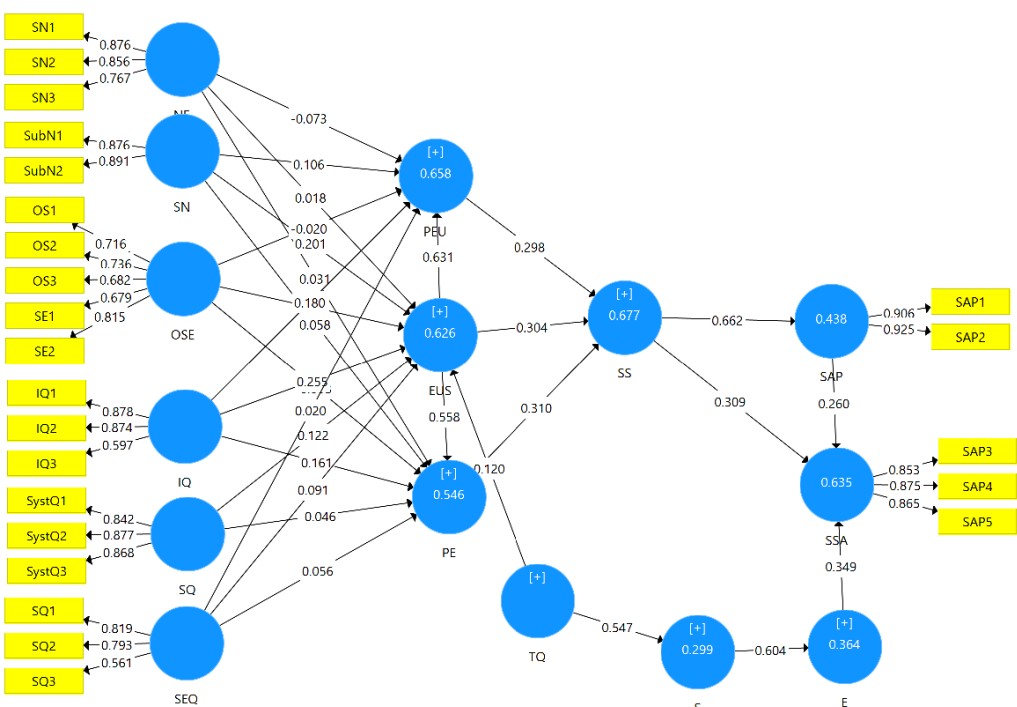

**Figure 3.** Structural model path coefficients.

**Table 5.** Results of the hypothesis testing and path analysis.

| H | Path | B-Coefficient | Standard Deviation | T Statistics | *p*-Values | Support |
|---|---|---|---|---|---|---|
| **H1a-1** | **SN -> EUS** | 0.106 | 0.105 | 0.063 | 1.685 | **0.093** |
| **H1a-2** | **SN -> PEU** | 0.201 | 0.199 | 0.059 | 3.420 | **0.001** |
| **H1a-3** | **SN- > PE** | 0.058 | 0.051 | 0.073 | 0.794 | **0.428** |
| **H1b-1** | **NE -> PEU** | −0.073 | −0.072 | 0.045 | 1.640 | **0.102** |
| **H1b-2** | **NE-> EUS** | 0.018 | 0.022 | 0.049 | 0.374 | **0.709** |
| **H1b-3** | **NE-> PE** | 0.031 | 0.034 | 0.056 | 0.549 | **0.583** |
| **H2a-1** | **OSE -> EUS** | 0.211 | 0.207 | 0.059 | 3.607 | **0.000** |
| **H2a-2** | **OSE -> PE** | 0.008 | 0.006 | 0.071 | 0.107 | **0.915** |
| **H2a-3** | **OSE -> PEU** | −0.020 | −0.027 | 0.063 | 0.317 | **0.751** |
| **H2b-1** | **S -> E** | 0.604 | 0.608 | 0.044 | 13.840 | **0.000** |
| **H2c-1** | **E -> SAP** | 0.349 | 0.349 | 0.057 | 6.099 | **0.000** |
| **H3a-1** | **SQ -> EUS** | 0.122 | 0.125 | 0.059 | 2.059 | **0.040** |
| **H3a-2** | **SQ -> PE** | −0.046 | −0.044 | 0.067 | 0.696 | **0.487** |
| **H3b-1** | **SEQ -> EUS** | 0.091 | 0.092 | 0.063 | 1.448 | **0.148** |
| **H3b-2** | **SEQ -> PE** | 0.056 | 0.055 | 0.064 | 0.877 | **0.381** |
| **H3b-3** | **SEQ -> PEU** | 0.020 | 0.020 | 0.052 | 0.379 | **0.705** |
| **H3c-1** | **IQ -> EUS** | 0.255 | 0.257 | 0.052 | 4.919 | **0.000** |
| **H3c-2** | **IQ -> PE** | 0.161 | 0.163 | 0.067 | 2.392 | **0.017** |
| **H3c-3** | **IQ -> PEU** | 0.180 | 0.180 | 0.066 | 2.752 | **0.006** |

**Table 5.** *Cont.*

| H | Path | B-Coefficient | Standard Deviation | T Statistics | *p*-Values | Support |
|---|------|---------------|--------------------|--------------|------------|---------|
| H4a-1 | EUS -> PEU | 0.631 | 0.635 | 0.072 | 8.778 | **0.000** |
| H4a-2 | EUS -> PE | 0.558 | 0.559 | 0.074 | 7.586 | **0.000** |
| H4a-3 | EUS -> SS | 0.304 | 0.309 | 0.084 | 3.639 | **0.000** |
| H4b-1 | PE -> SS | 0.310 | 0.309 | 0.067 | 4.666 | **0.000** |
| H4c-1 | PEU -> SS | 0.298 | 0.297 | 0.069 | 4.309 | **0.000** |
| H5a-1 | TQ -> EUS | 0.120 | 0.123 | 0.065 | 1.855 | **0.064** |
| H5a-2 | TQ -> S | 0.547 | 0.553 | 0.050 | 10.849 | **0.000** |
| H6a-1 | SS -> SAP | 0.662 | 0.662 | 0.040 | 16.391 | **0.000** |
| H6a-2 | SS -> SSA | 0.309 | 0.309 | 0.059 | 5.248 | **0.000** |
| H6b-1 | SAP -> SSA | 0.260 | 0.262 | 0.069 | 3.772 | **0.000** |

Three constructs, PEU, USE, and PE, were the main determinants of student satisfaction, together explaining 67.7% of the variance.

Fourth, we assessed the predictive relevance denounced as $Q^2$ during the blindfolding in SmartPLS. If the model performs a predictive relevance (values of $Q^2$ higher than 0), the test will demonstrate accuracy in predicting items' data points [94]. The authors established that a value of $Q^2$ of 0.02 denotes a small predictive relevance, a value of 0.15 shows a medium relevance, and a value of 0.35 presents a large predictive relevance. Table 6 shows the $Q^2$ of the endogenous variables, with five with strong prediction power (PE, PEU, SS, SSA, and SAP) and three with moderate prediction power (USE, S, and E).

**Table 6.** $Q^2$ results, showing the predictive relevance.

|  | $Q^2$ | Predictive Relevance |
|---|-------|----------------------|
| E | 0.185 | Moderate |
| EUS | 0.301 | Moderate |
| PE | 0.420 | Large |
| PEU | 0.520 | Large |
| S | 0.157 | Moderate |
| SAP | 0.359 | Large |
| SS | 0.463 | Large |
| SSA | 0.467 | Large |

Results suggest that the model has considerable predictive power due to the value of $Q^2$ for student academic performance (SAP) and student academic achievement (SSA).

Finally, the last step was to assess the model fit, as proposed [110]; that is, how well the specified model represents the underlying theory [111]. Ref. [112] proposed a set of fit measures, but stated that they have been introduced to provide a comparison to CB-SEM results rather than to represent an appropriate PLS-SEM index:

1.  Standardized Root Mean Square Residual (SRMR), which is an absolute measure of model fit proposed to prevent misspecification of the model [104]. A value less than 0.10 or 0.08 (a more conservative version, see [113]) is considered a good fit. The SRMR for this study is 0.07, which is below the lower cut-off value suggested in the literature.

2.  Root Mean Square Residual (RMS$_{theta}$) verifies "the degree to which the outer model residuals correlate" [104]. Closeness to 0 indicates a good model fit ($\leq 0.12$ to indicate

a good model fit) [101,104]. Using SmartPLS, the value of RMS$_{theta}$ is 0.116 which indicates a good model fit.

3. Normed Fit Index (NFI), which provides an incremental fit measure. Therefore, one of the main disadvantages is that it does not penalize for model complexity; i.e., the more parameters in the model, then the larger (i.e., better) the NFI result. Closeness of the NFI to 1 indicates a better fit. NFI values above 0.9 usually represent acceptable fit [110]. In our case, the value of NFI is 0.613.

4. Finally, the model's goodness of fit (GoF) is defined as "how well the specified model reproduces the observed covariance matrix among the indicator items" [105] and this is our last criterion to assess the overall model fit. The purpose of GoF is to account for the model at both levels; i.e., the measurement and the structural models, with a focus on the overall performance [114]. There is no measure of global fit in PLS. However, investigators have suggested a global GoF, which is defined as the geometric mean of the average communality and average $R^2$ of the endogenous constructs [115]. The GoF cut-off values used in this study were proposed by [116]:

- GoF less than 0.1 means no fit;
- GoF between 0.1 and 0.25 means small fit;
- GoF between 0.25 and 0.36 means medium;
- GoF greater than 0.36 means large.

The model's goodness of fit for this research is 0.599, which means a large overall performance; in fact, it is significantly above the threshold value that constitutes a large fit.

## 7. Discussion

This study aimed to analyze how different factors can predict students' accuracy in self-assessing their accomplishments within the secondary student population.

To explore the factors predicting student academic achievement (SSA) during COVID-19, a new model (3S-T model) was developed, based on extended TAM, ISSM, EIAM, and OSE models. This 3S-T model was used to explain the secondary school student perceptions of the process of the adoption of E-learning during the lockdown. The complicated age of these students and the lack of studies focused on this crucial educational level highlight the need of the present analysis. In this vein, our approach is strongly linked to Goal 4 of the Agenda adopted by the United Nations [117], which pursues Quality Education in order to achieve sustainable development among countries.

Hypothesis H1a-1, H2a-1, H2b-1, H2c-1, H3b-1, H3c-1, and H2a gained empirical support. Thus, aspects related to subject norm, one-self efficacy, strategy, engagement, and information. System quality, understood as the available information at the E-learning system, is helpful. The interest and reliability of the information available is an important aspect in order to contribute to the general satisfaction and perceived usefulness of the E-learning system. Furthermore, several aspects related to the students' perception of the system quality, in particular, the ones concerning the site, such as the easiness to understand navigation, easiness to find the information, and to have a good website structure, are also vital aspects to EUS. These results support that information quality and system quality are determinants of the perceived satisfaction and the perceived usefulness.

In this study, it was shown that engagement profiles and study strategies are important predictors of SSA. This finding is consistent with other research where engagement have been related, such as the studies of [51] and [118].

Contrary to our prediction, H3a1, H3a-2, and H3a-3 were rejected, and this is interpreted as (i) their age, as the current students were born using the internet and platforms/systems of all types and complexity; (ii) in the survey, 17.5% affirmed that they do not have computer skills, which can be considered a "false-negative" answer because all the students in the educational system of Spain study informatics, and in the more experienced centers, all students use digital books to learn; (iii) developers of E-learning platforms/systems (most of them learned how to handle platforms when they were older) underestimate the skills of students for navigation, for searching information, and for

understanding a webpage structure; and (iv) authors who assume that these hypotheses will have a positive effect are based on the beginning of the internet era, with students and tutors with limited skills and strategies.

Statistical analysis established that there are positive relationships among EUS and PEU (H4a-1), EUS and PE (H4a-2), and EUS and SS (H4a-3). These results could suggest that if students increase the usefulness towards sustainability, they may potentially increase their perceived usefulness and perceived enjoyment. Therefore, students considered that E-learning during the lockdown was valuable, and it creates a suitable atmosphere to learn, affecting positively the E-learning performance and effectiveness as well their control and utility. Besides, this mentioned E-learning atmosphere made the E-learning more enjoyable, pleasant, and funny, improving the general student satisfaction, using it again if necessary.

H5a-1 and H5a-2 were supported, since the instructor/tutor is the key in an E-learning environment [51], especially with underage students who are more dependent in every way.

H6a-1 and H6b-1 were also supported. More satisfied students gain more benefits, impacting their learning achievements. These results are consistent and in the line with previous research [10,35,51,119,120]. Obviously, students who feel satisfied have enhanced performance.

H6a-2 was also supported. Students were satisfied with E-learning and willing to use it again if necessary, and they also found what they need, which allowed them to achieve educational and personal goals, improving their creativity, their knowledge and information, and their experiences and performance.

Finally, H4b-1 and H4c-1 were strongly supported. Thus, it means that perceived usefulness and perceived enjoyment are determinants of student satisfaction. Therefore, students who perceived that E-learning is useful and can be used as a form of enjoyment were successful.

The results show that benefits of the 3S-T model are achieved; thus, the use of E-learning increases the learning performance and learning achievements. Our obtained results using the 3S-T model are in the line with other studies [35,51,102,119].

## 8. Conclusions

In the complicated period of the COVID-19 pandemic, and during the global lockdown, E-learning was the only resource capable of replacing traditional in-person learning procedures. Surprisingly, the age of secondary students being the most complicated in all terms (educational, behavior, and social), there is a great lack of studies that analyze the E-learning process during the global lockdown at this crucial educational level.

In the current research work, we have developed an original model 3S-T based on the current theoretical framework in order to identify a range of success predictors and to measure the success of the E-learning model based on a questionnaire answered by more than 200 students.

The measurement and success of the different factors that influence the E-learning were evaluated using the mentioned 3S-T model. In this way, the initial objective was accomplished, and we found that factors related to subject norm, one-self efficacy, strategy engagement and information, system quality, interest and reliability of the information available, and factors related to the student perception of the system quality are vital and similar to the results of other authors.

This article contributes to the emerging literature related to the analysis of E-learning systems success, providing a comprehensive multidimensional model that takes into account the main dimensions and subdimensions of four approaches: ISSM, TAM, ElAM, and OLSE.

The methodological procedure employed the regression technique of PLS-SEM using the SmartPLS 3.6 software. According to the results, the use of E-learning increased the learning performance and the learning achievements of the secondary students during the global lockdown. The adoption of the present 3S-T model to analyze and to measure the success of E-learning during the COVID pandemic is key. In this vein, the 3S-T

model should be extended to other educational levels, such as primary education or at the university level, in futures articles.

**Supplementary Materials:** The following supporting information can be downloaded at: https://www.mdpi.com/article/10.3390/su142013261/s1 (answers from the students).

**Author Contributions:** M.M.-G. and E.B. conceived and designed the experiments; E.B. performed the experiments and M.M.-G. and C.B.-E. analyzed the data; M.M.-G. and E.B. contributed analysis tools; C.B.-E., M.M.-G. and E.B. wrote the paper. E.B. and C.B.-E. revised the paper. All authors have read and agreed to the published version of the manuscript.

**Funding:** This research received no external funding.

**Informed Consent Statement:** Informed consent was obtained from all subjects involved in the study.

**Data Availability Statement:** The authors confirm that the data supporting the findings of this study are available within the article and its Supplementary Materials.

**Conflicts of Interest:** The authors declare no conflict of interest.

## Abbreviations

| | |
|---|---|
| 3S-T | Newly proposed model (social aspects, student factors, system factors, and tutor capabilities) |
| 5Qs | Model of the 5 Qualities (object, process, infrastructure, interaction, and communication atmosphere) |
| ASS | Student Learning Achievements |
| AVE | Average Variance Extracted |
| CB-SEM | Covariance-Based and Structural Equation Modelling |
| CFA | Confirmatory Analysis |
| COU | University Orientation Course (Curso de Orientación Universitaria) |
| COVID-19 | Corona Virus Disease 2019 |
| CR | Composite Reliability |
| DTPB | Decomposed Theory of Planned Behavior |
| E | Engagement |
| EIAM | E-learning Self-Acceptance Measure |
| ELQ | E-Learning Quality |
| ELS | E-Learner Satisfaction |
| ELSS | E-Learning System Success |
| EESS | Evaluating E-learning Systems Success |
| ESO | Compulsory Secondary School (Educación Secundaria Obligatoria) |
| EUS | Intention to Use |
| GoF | Goodness of Fit |
| HTMT | Heterotrait–Monotrait ratio |
| ICT | Information and Communication Technologies |
| IQ | Information Quality |
| ISS | Information Systems Success |
| ISSM | Information Systems Success Model |
| IT | Information Technologies |
| LMS | Learning Management Systems |
| NE | Social Net working |
| OLE | Online Learning Environment |
| OLSE | Online learning self-efficacy |
| OSE | One-Self Efficacy |
| PE | Perceived Enjoyment |
| PEU | Perceived Usefulness |
| PLS | Partial Least Square |

| PLS-SEM | Partial Least Square and Structural Equation Model |
|---|---|
| $Q^2$ | Goodness of PLS prediction |
| QoI | Quality of Interaction |
| $R^2$ | Determination Coefficient |
| RMS | Root Mean Square |
| S | Strategy |
| SAP | Student Academic Performance |
| SD | Standard Deviation |
| SEM | Structural Equation Model |
| SmartPLS | Commercial Computational Code |
| SN | Subjective Norm |
| SERVQUAL | Multiple-item scale for measuring consumer perceptions of Service Quality |
| SEQ | Service Quality |
| SQ | System Quality |
| SRMR | Standardized Root Mean Residual |
| SS | Student Satisfaction |
| SSA | Student Self-Assessment |
| TAM | Technology Acceptance Model |
| TAM2 and 3 | Extensions of the TAM model |
| TRA | Theory of Reasoned Action |
| TT | Tutor Quality |
| USM | User Satisfaction Model |
| UTAUT | Unified Theory of Acceptance and Use of Technology |
| UTAUT2 | Modification of UTAUT model |
| VIF | Variance Inflation Factor |

## Appendix A

**Table A1.** Questionnaire.

| | | Code | Related Studies |
|---|---|---|---|
| **Information Quality** | The information at the E-learning system available is helpful | IQ1 | [42,72] |
| | The information available is interesting | IQ2 | |
| | The information available is reliable | IQ3 | |
| **Service Quality** | The E-learning has a mechanism for overcoming the problems that I am facing quickly | SEQ1 | [10,20,24,38,42] |
| | The system on E-learning site is up to date. | SEQ2 | |
| | You feel safe with the E-learning system in terms of security and privacy | SEQ3 | |
| **System Quality** | The E-learning site has easy-to-understand navigation | SQ1 | [16,20,22,42,68] |
| | The E-learning site allows me to find the information I need easily | SQ2 | |
| | The E-learning site has a good website structure | SQ3 | |
| **Intention to Use** | I use the E-learning site to find information | USE1 | [38,39,72,82] |
| | I use E-learning to assess my skills | USE2 | |
| | I use E-learning to increase the chances of achieving better results | USE3 | |
| **Perceived Usefulness** | Using the E-learning system improves my learning performance. | PEU1 | [38,46,70,83,89,90] |
| | Using the E-learning system enhances my learning effectiveness | PEU2 | |
| | Using the E-learning system gives me greater control over learning. | PEU3 | |
| | I find the E-learning system to be useful in my learning | PEU4 | |

**Table A1.** *Cont.*

|  |  | Code | Related Studies |
|---|---|---|---|
| **Perceived Enjoyment** | I find using the E-learning system to be enjoyable. | PE1 | [10,18,42,43] |
|  | The actual process of using the E-learning system is pleasant. | PE2 |  |
|  | I have fun using the E-learning system. | PE3 |  |
| **Student Satisfaction** | If there is any chance to use online learning again, I will gladly do it. | SS1 | [9,21,22,24,27,79] |
|  | I am satisfied with the E-learning process | SS2 |  |
|  | I feel online learning gives me what I need | SS3 |  |
| **Use as Sustainability** | I spend a lot of time exploring within the E-learning system. | EUS1 | [22,43,62,66,75] |
|  | I believe that the use of the system is valuable | EUS2 |  |
|  | E-learning provide suitable learning environment | EUS3 |  |
|  | I think that using E-learning is well suited for the way to learn. | EUS4 |  |
| **Social Networking** | I enjoy my time when using social networking tools. | EN1 | [19,38,63] |
|  | Social networking tools increase students' creativity and interactivity. | EN2 |  |
|  | Social networking tools facilitate knowledge sharing. | EN3 |  |
| **Student Learning Achievements** | Achieving educational goals | ASS1 | [22,70,73] |
|  | Achieving personal goals | ASS2 |  |
|  | I feel the E-learning system helps me improve my creativity. | ASS3 |  |
|  | I feel the E-learning system helps me improve my knowledge and information. | ASS4 |  |
|  | I feel the E-learning system helps me improve my experiences and performance | ASS5 |  |
| **Subjective Norm** | My teacher is very supportive of online learning system use for my learning | SN1 | [59–61,87,88] |
|  | The management of my university support the E-learning activities | SN2 |  |
| **Tutor Quality** | My tutor could explain the concepts clearly | TQ1 | [46,49,85,88,92] |
|  | My tutor was knowledgeable in ICT | TQ2 |  |
|  | My tutor was focused on helping me to learn | TQ3 |  |
|  | The tutorial activities were well-manage | TQ4 |  |
|  | My tutor was accessible when I needed to consult them | TQ5 |  |
|  | My tutor was patient when they interacted with me | TQ6 |  |
|  | The group sessions were well facilitated | TQ7 |  |
| **Self-efficacy** | I am willing to accept the challenge | OSE1 | [38,59,67–69,71] |
|  | I am sure that I can complete all the stages that exist on E-learning site well. | OSE2 |  |
| **Online Self-Efficacy** | I am sure that I can use synchronous technology to communicate with others (such as Skype) | OS1 | [11,46–49,54,91] |
|  | I am sure that I can manage time effectively and complete all assignments on time | OS2 |  |
|  | I am sure that I can learn without being in the same room as the instructor and other students | OS3 |  |

**Table A1.** *Cont.*

| | | | Code | Related Studies |
|---|---|---|---|---|
| **Engagement** | | I feel strong and vigorous when I am studying or going to E-learning classes. | E1 | [2,10,51,56,72] |
| | | When the day starts I feel like going to class or studying | E2 | |
| | | I am enthused about my studies | E3 | |
| | | My studies inspire me to do new things | E4 | |
| | | I am proud of doing this career | E5 | |
| | | I am happy when I am doing tasks related to my studies | E6 | |
| | | I am involved in my studies | E7 | |
| **Strategy** | | I tend to plan the time I am going to spend studying | S1 | [36,49,68,80] |
| | | I start studying from the beginning of the course | S2 | |
| | | I take notes of the teachers' explanations. | S3 | |
| | | I expand the information with complementary bibliography. | S4 | |
| | | I have difficulties in following the teacher's explanations in E-learning class. | S5 | |
| | | I make outlines of the material I am going to study. | S6 | |
| | | When I study for an exam I think of questions that can be included in the exam. | S7 | |

**Table A2.** Table of the outer loadings.

| | EUS | z | EN | IQ | OSE | PE | PEU | SEQ | SE | S | SAP | ASS | SS | SN | SQ | TQ |
|---|---|---|---|---|---|---|---|---|---|---|---|---|---|---|---|---|
| **E1** | | | 0.71 | | | | | | | | | | | | | |
| **E2** | | | 0.74 | | | | | | | | | | | | | |
| **E3** | | | 0.80 | | | | | | | | | | | | | |
| **E4** | | | 0.82 | | | | | | | | | | | | | |
| **E5** | | | 0.65 | | | | | | | | | | | | | |
| **E6** | | | 0.76 | | | | | | | | | | | | | |
| **E7** | | | 0.56 | | | | | | | | | | | | | |
| **EUS1** | 0.72 | | | | | | | | | | | | | | | |
| **EUS2** | 0.73 | | | | | | | | | | | | | | | |
| **EUS3** | 0.78 | | | | | | | | | | | | | | | |
| **EUS4** | 0.80 | | | | | | | | | | | | | | | |
| **USE1** | 0.64 | | | | | | | | | | | | | | | |
| **USE2** | 0.60 | | | | | | | | | | | | | | | |
| **USE3** | 0.65 | | | | | | | | | | | | | | | |
| **IQ1** | | | | 0.88 | | | | | | | | | | | | |
| **IQ2** | | | | 0.87 | | | | | | | | | | | | |
| **IQ3** | | | | 0.60 | | | | | | | | | | | | |
| **OS1** | | | | | 0.72 | | | | | | | | | | | |
| **OS2** | | | | | 0.74 | | | | | | | | | | | |
| **OS3** | | | | | 0.68 | | | | | | | | | | | |

**Table A2.** *Cont.*

| | EUS | z | EN | IQ | OSE | PE | PEU | SEQ | SE | S | SAP | ASS | SS | SN | SQ | TQ |
|---|---|---|---|---|---|---|---|---|---|---|---|---|---|---|---|---|
| **PE1** | | | | | | 0.90 | | | | | | | | | | |
| **PE2** | | | | | | 0.87 | | | | | | | | | | |
| **PE3** | | | | | | 0.91 | | | | | | | | | | |
| **PEU1** | | | | | | | 0.88 | | | | | | | | | |
| **PEU2** | | | | | | | 0.93 | | | | | | | | | |
| **PEU3** | | | | | | | 0.90 | | | | | | | | | |
| **S1** | | | | | | | | | | 0.83 | | | | | | |
| **S2** | | | | | | | | | | 0.82 | | | | | | |
| **S3** | | | | | | | | | | 0.69 | | | | | | |
| **S4** | | | | | | | | | | 0.75 | | | | | | |
| **S6** | | | | | | | | | | 0.57 | | | | | | |
| **SAP1** | | | | | | | | | | | | 0.91 | | | | |
| **SAP2** | | | | | | | | | | | | 0.92 | | | | |
| **SAP3** | | | | | | | | | | | 0.85 | | | | | |
| **SAP4** | | | | | | | | | | | 0.88 | | | | | |
| **SAP5** | | | | | | | | | | | 0.87 | | | | | |
| **SE1** | | | | | 0.68 | | | | | | | | | | | |
| **SE2** | | | | | 0.82 | | | | | | | | | | | |
| **SN1** | | | | | | | | | 0.88 | | | | | | | |
| **SN2** | | | | | | | | | 0.86 | | | | | | | |
| **SN3** | | | | | | | | | 0.77 | | | | | | | |
| **SQ1** | | | | | | | | 0.82 | | | | | | | | |
| **SQ2** | | | | | | | | 0.79 | | | | | | | | |
| **SQ3** | | | | | | | | 0.56 | | | | | | | | |
| **SS1** | | | | | | | | | | | | | 0.79 | | | |
| **SS2** | | | | | | | | | | | | | 0.87 | | | |
| **SS3** | | | | | | | | | | | | | 0.85 | | | |
| **SubN1** | | | | | | | | | | | | | | 0.88 | | |
| **SubN2** | | | | | | | | | | | | | | 0.89 | | |
| **SystQ1** | | | | | | | | | | | | | | | 0.84 | |
| **SystQ2** | | | | | | | | | | | | | | | 0.88 | |
| **SystQ3** | | | | | | | | | | | | | | | 0.87 | |
| **TQ1** | | | | | | | | | | | | | | | | 0.78 |
| **TQ2** | | | | | | | | | | | | | | | | 0.79 |
| **TQ3** | | | | | | | | | | | | | | | | 0.82 |
| **TQ4** | | | | | | | | | | | | | | | | 0.85 |
| **TQ5** | | | | | | | | | | | | | | | | 0.75 |
| **TQ6** | | | | | | | | | | | | | | | | 0.74 |
| **TQ7** | | | | | | | | | | | | | | | | 0.72 |

**Table A3.** Table of the significance of the outer loadings.

| | Original Sample (O) | Sample Mean (M) | Standard Deviation (STDEV) | T Statistics (|O/STDEV|) | *p* Values |
|---|---|---|---|---|---|
| E1 ← E | 0.712 | 0.713 | 0.041 | 17.559 | 0.000 |
| E2 ← E | 0.740 | 0.734 | 0.039 | 18.899 | 0.000 |
| E3 ← E | 0.798 | 0.795 | 0.028 | 28.108 | 0.000 |
| E4 ← E | 0.817 | 0.815 | 0.025 | 33.138 | 0.000 |
| E5 ← E | 0.650 | 0.649 | 0.045 | 14.366 | 0.000 |
| E6 ← E | 0.762 | 0.760 | 0.040 | 19.000 | 0.000 |
| E7 ← E | 0.564 | 0.563 | 0.057 | 9.944 | 0.000 |
| EUS1 ← EUS | 0.716 | 0.716 | 0.037 | 19.377 | 0.000 |
| EUS2 ← EUS | 0.730 | 0.727 | 0.037 | 19.817 | 0.000 |
| EUS3 ← EUS | 0.784 | 0.785 | 0.029 | 27.255 | 0.000 |
| EUS4 ← EUS | 0.803 | 0.802 | 0.026 | 30.353 | 0.000 |
| IQ1 ← IQ | 0.878 | 0.876 | 0.020 | 42.881 | 0.000 |
| IQ2 ← IQ | 0.874 | 0.874 | 0.016 | 53.332 | 0.000 |
| IQ3 ← IQ | 0.597 | 0.590 | 0.078 | 7.603 | 0.000 |
| OS1 ← OSE | 0.717 | 0.709 | 0.053 | 13.563 | 0.000 |
| OS2 ← OSE | 0.736 | 0.735 | 0.043 | 16.966 | 0.000 |
| OS3 ← OSE | 0.682 | 0.679 | 0.040 | 17.223 | 0.000 |
| PE1 ← PE | 0.898 | 0.897 | 0.015 | 61.367 | 0.000 |
| PE2 ← PE | 0.876 | 0.875 | 0.024 | 36.351 | 0.000 |
| PE3 ← PE | 0.907 | 0.906 | 0.015 | 62.369 | 0.000 |
| PEU1 ← PEU | 0.882 | 0.882 | 0.018 | 48.959 | 0.000 |
| PEU2 ← PEU | 0.928 | 0.928 | 0.016 | 59.860 | 0.000 |
| PEU3 ← PEU | 0.901 | 0.900 | 0.015 | 61.628 | 0.000 |
| S1 ← S | 0.831 | 0.831 | 0.020 | 41.547 | 0.000 |
| S2 ← S | 0.823 | 0.821 | 0.024 | 34.328 | 0.000 |
| S3 ← S | 0.692 | 0.690 | 0.047 | 14.674 | 0.000 |
| S4 ← S | 0.752 | 0.751 | 0.036 | 20.767 | 0.000 |
| S6 ← S | 0.574 | 0.571 | 0.062 | 9.324 | 0.000 |
| SAP1 ← SSA | 0.906 | 0.905 | 0.018 | 49.304 | 0.000 |
| SAP2 ← SSA | 0.925 | 0.924 | 0.012 | 77.429 | 0.000 |
| SAP3 ← SAP | 0.847 | 0.847 | 0.022 | 38.066 | 0.000 |
| SAP4 ← SAP | 0.878 | 0.876 | 0.023 | 38.610 | 0.000 |
| SAP5 <- SAP | 0.869 | 0.868 | 0.020 | 43.472 | 0.000 |
| SE1 ← OSE | 0.679 | 0.677 | 0.052 | 13.044 | 0.000 |
| SE2 ← OSE | 0.815 | 0.812 | 0.022 | 37.012 | 0.000 |
| SN1 ← SN | 0.876 | 0.875 | 0.023 | 38.352 | 0.000 |
| SN2 ← SN | 0.856 | 0.854 | 0.031 | 27.748 | 0.000 |
| SN3 ← SN | 0.767 | 0.761 | 0.055 | 14.064 | 0.000 |
| SQ1 ← SQ | 0.819 | 0.817 | 0.037 | 22.408 | 0.000 |
| SQ2 ← SQ | 0.794 | 0.789 | 0.040 | 20.005 | 0.000 |
| SQ3 ← SQ | 0.561 | 0.553 | 0.082 | 6.878 | 0.000 |
| SS1 ← SS | 0.781 | 0.780 | 0.037 | 21.303 | 0.000 |
| SS2 ← SS | 0.872 | 0.872 | 0.020 | 43.249 | 0.000 |
| SS3 ← SS | 0.854 | 0.853 | 0.017 | 50.325 | 0.000 |
| SubN1 ← SubNorm | 0.876 | 0.875 | 0.022 | 40.381 | 0.000 |
| SubN2 ← SubNorm | 0.891 | 0.891 | 0.017 | 52.230 | 0.000 |
| SystQ1 ← SystQ | 0.840 | 0.838 | 0.029 | 29.456 | 0.000 |
| SystQ2 ← SystQ | 0.879 | 0.880 | 0.018 | 48.935 | 0.000 |
| SystQ3 ← SystQ | 0.867 | 0.867 | 0.024 | 36.746 | 0.000 |
| TQ1 ← TQ | 0.781 | 0.779 | 0.031 | 25.127 | 0.000 |
| TQ2 ← TQ | 0.785 | 0.783 | 0.031 | 25.467 | 0.000 |
| TQ3 ← TQ | 0.822 | 0.818 | 0.029 | 28.402 | 0.000 |
| TQ4 ← TQ | 0.848 | 0.846 | 0.022 | 38.970 | 0.000 |
| TQ5 ← TQ | 0.753 | 0.751 | 0.045 | 16.612 | 0.000 |
| TQ6 ← TQ | 0.745 | 0.739 | 0.042 | 17.635 | 0.000 |
| TQ7 ← TQ | 0.718 | 0.714 | 0.041 | 17.584 | 0.000 |
| USE1 ← EUS | 0.642 | 0.636 | 0.050 | 12.707 | 0.000 |
| USE2 ← EUS | 0.601 | 0.596 | 0.056 | 10.705 | 0.000 |
| USE3 ← EUS | 0.654 | 0.650 | 0.050 | 13.067 | 0.000 |

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
