# Peer review of "Development and Validation of an E-Learning Education Model in the COVID-19 Pandemic: A Case Study in Secondary Education"

_sustainability, doi:10.3390/su142013261_

Round 1

Reviewer 1 Report

The paper is a good quality. I miss one information:

The survey was done among students. Why not among educators? They should also contribute to e-learning quality. You do not need to perform additional research. Just explain your reasons and, if possible, refer to relevant resources.

It seems to me that the process on Figure 1 leads to two outputs. In my opinion, there should be independent i.e. there should not be any arrow between them. Delete it or add your explanation.

There are also minor errors:

Line 159: "information quality" appears 2 times.

Line 186: Service Quality (Capital Q)

Line 203: "construct" or "constructs"?

Title on Line 289 begins with dot "."

Author Response

REFEREE #1

First of all, we want to thank the reviewer for their useful comments and suggestions. For this reason, all his comments and suggestions have been incorporated in the present revised version of the manuscript. We think all of them have noticeably improved the original draft of the manuscript. In the following we explain point by point how these suggestions have been incorporated in the revised version.

The paper is a good quality. I miss one information:

The survey was done among students. Why not among educators? They should also contribute to e-learning quality. You do not need to perform additional research. Just explain your reasons and, if possible, refer to relevant resources.

ANSWER: We appreciate the reviewer’s suggestion.

The researchers focused the study on Secondary Education students because they have had the greatest impact of E-learning. They hardly had access to internet platforms or electronic devices, so it has been a great effort for them to continue with their studies during the pandemic, especially in the most disadvantaged households.

However, we are aware that it has also been difficult for educators but there are many papers that reveal a positive finding revealed a positive relationship between teacher`s perception and technology acceptance (eLearning) during Covid-19 (Alhumaid et al., 2020; Almanthari et al., 2020, Cheok et al. 2017, Mahdizadeh et al., 2008) so we have updated the text in the introduction and in references as follows:

“The measurement/estimation of the success of E-learning initiatives has been vastly investigated [2, 3, 13]. A quick review of the literature shows that different studies use different theoretical frameworks, such as the Technology Acceptance Model (TAM) [14, 15], Information Systems Success (ISS) [9], SERVQUAL [16], the Decomposed Theory of Planned Behavior (DTPB) [17], the UTAUT and UTAUT2 models [18, 19] and the 5Q model [20]. In addition, many E-learning success and quality evaluation models have been proposed, such as the E-Learning System Success (ELSS), the Evaluating E-learning Systems Success evaluation (EESS), the E-Learning Quality (ELQ), the E-learner Satisfaction (ELS) and the User Satisfaction Model (USM) [21, 22, 23, 24, 25, 26, 27]. In the same way, along with these researchers many different dimensions, factors or constructs have been considered to evaluate the E-learning performance in each particular application. So, various factors have been identified for the success of E-learning. In this paper a comprehensive model is presented for measuring the success of E-learning systems on student academic performance and on students learning achievements. We are aware that it has also been difficult for educators but there are many papers that reveal a positive relationship between teacher`s perception and technology acceptance (E-Learning) [28-31]. To this end, the fundamental purpose of this study is the development and validation of a tool that provides data to help better understand the factors that influence E-learning, and not only this, but also to be able to estimate the importance of each of them”.

It seems to me that the process on Figure 1 leads to two outputs. In my opinion, there should be independent i.e. there should not be any arrow between them. Delete it or add your explanation.

ANSWER: We again appreciate the reviewer’s suggestion, however in our model, we have assumed that student satisfaction is a determinant of the benefits construct, it is to say students learning achievements (SSA). Therefore, we assume the following hypothesis:

H6a-1: Student Satisfaction (SS) toward the E-learning system positively influences students' academic performance (SAP).

H6a-2: Student Satisfaction (SS) toward the E-learning system positively influences students' learning achievement (SSA).

H6b-1: Students' academic performance (SAP), positively influences students' learning achievement (SSA).

And the three hypotheses were supported according to the statistic p-value.

There are also minor errors:

Line 159: "information quality" appears 2 times.

ANSWER: We agree with the referee suggestion. So, we have checked it and we have delete “Information quality”.

Line 186: Service Quality (Capital Q)

ANSWER: We again agree with the referee suggestion. So, we have checked it and we have change into capital letter Q.

Line 203: "construct" or "constructs"?

ANSWER: We appreciate the reviewer’s suggestion. We have changed “constructs” by “construct”.

Title on Line 289 begins with dot "."

ANSWER: We appreciate the reviewer’s suggestion. We have delated the dot.

English should be checked once more

ANSWER: We agree the suggestion made by the reviewer. We have revised the full text, in order to improve the English language and style.

Reviewer 2 Report

Greetings to the author. Congratulations, you have succeeded in researching and writing articles related to E-Learning. I thank you for the opportunity to review your article. After my review, there are several parts that need to be re-explained, including:

Line 12: After reading the contents of this article, we did not find the abbreviation of this abbreviation (3S-T), preferably in an article the abbreviation of an abbreviation is written at the beginning or on the first page. Please lengthen any words from this abbreviation.

Line 134: If this theoretical framework can be arranged in the form of a flow chart, it will be very helpful for readers to understand the contents of this article holistically, including the models adopted and the constructs used in measuring the E-Learning education model, as well as the stages in the development of the E-Learning education model.

Line 152: It is explained in detail whether the models used are only to measure each construct in the E-Learning education model or also to explain the stages of model development.

Line 277: We recommend that the abbreviation for 3S-T appear at the beginning of the page on the first page so that readers can more easily understand the descriptions related to 3S-T on the pages.

Line 455: In this section, the method is explained systematically starting from the research approach used to the data analysis method used. This sequential description is important so that the reader can follow this study easily.

Line 683: If we refer to the description on the introductory page, one of the objectives of this research is to find out the factors that influence E-Learning. This goal has not been described in depth in this conclusion. Please add these objectives to this conclusion.

Author Response

REFEREE #2

First of all, we want to thank the reviewer for their useful comments and suggestions. For this reason, all his comments and suggestions have been incorporated in the present revised version of the manuscript. We think all of them have noticeably improved the original draft of the paper. In the following we explain point by point how these suggestions have been incorporated in the revised version of the manuscript.

Greetings to the author. Congratulations, you have succeeded in researching and writing articles related to E-Learning. I thank you for the opportunity to review your article. After my review, there are several parts that need to be re-explained, including:

Line 12: After reading the contents of this article, we did not find the abbreviation of this abbreviation (3S-T), preferably in an article the abbreviation of an abbreviation is written at the beginning or on the first page. Please lengthen any words from this abbreviation.

ANSWER: We agree with the referee suggestion. We have included a list of abbreviations before the introduction including the abbreviation of 3S-T.

Line 134: If this theoretical framework can be arranged in the form of a flow chart, it will be very helpful for readers to understand the contents of this article holistically, including the models adopted and the constructs used in measuring the E-Learning education model, as well as the stages in the development of the E-Learning education model.

ANSWER: We agree the suggestion made by the reviewer. We have illustrated in Figure 2 and update the text as follows:

We have illustrated in Figure 2 the two stages of the methodology. Stage 1 addresses the evaluation of reflective and formative measurement models or both (measurement (models examine the measurement theory). The Stage 2 addressed the valuation of structural model (that covers the structural theory that involves testing the proposed hypotheses and that addresses the relationships among the latent variables)

Line 152: It is explained in detail whether the models used are only to measure each construct in the E-Learning education model or also to explain the stages of model development.

ANSWER: We again agree with the referee suggestion. We have uploaded the text as follows:

“To this end, the fundamental purpose of this study is the development and validation of a tool that provides data to help better understand the factors that influence E-learning of students, and not only this, but also to be able to estimate the importance of each of them. In this model not only the measurement model for each construct has been validated, but also the relationships between the measurement model and the structural model have been determined”.

Line 277: We recommend that the abbreviation for 3S-T appear at the beginning of the page on the first page so that readers can more easily understand the descriptions related to 3S-T on the pages.

ANSWER: We agree with the referee suggestion. We have included a list of abbreviations before the introduction including the abbreviation for 3S-T.

Line 455: In this section, the method is explained systematically starting from the research approach used to the data analysis method used. This sequential description is important so that the reader can follow this study easily.

ANSWER: We again agree with the suggestion made by the reviewer. We have illustrated in Figure 2 and update the text as follows:

“We have illustrated in Figure 2 the two stages of the methodology. Stage 1 addresses the evaluation of reflective and formative measurement models or both (measurement (models examine the measurement theory). Stage 2 addressed the evaluation of structural model (that covers the structural theory that involves testing the proposed hypotheses and that addresses the relationships among the latent variables) [97].”

Figure 2. PLS-SEM model evaluation (Adapted from [98])

Reference

[97] Hair Jr., J.F.; Hult, G. T.M.; Ringle, C.; Sarstedt, M. A primer on partial least squares structural equation modeling (PLS-SEM). 2nd Edition 2017. Sage Publications.

[98] Sarstedt, M.; Ringle, C.M.; Hair, J. Partial Least Squares Structural Equation Modeling. Handbook of Market Research 2017, Heidelberg, Germany, Springer Publisher.

Line 683: If we refer to the description on the introductory page, one of the objectives of this research is to find out the factors that influence E-Learning. This goal has not been described in depth in this conclusion. Please add these objectives to this conclusion.

ANSWER: We again agree with the referee suggestion. We have uploaded the text as follows:

“The measurement and success of the different factors that influence the E-learning are evaluated using the mentioned 3S-T model. In this way, the initial objective is accomplished and we find that factors related to Subject Norm, One-Self Efficacy, Strategy Engagement and Information, System quality, interest and reliability of the information available and factors related to the student perception of the System Quality are vital and in similar trend of results of other authors.”

Reviewer 3 Report

Researchers should expand their knowledge Alternatively, there are scholars' references on determining the ratio of samples to variables for SEM analysis.

Author Response

REFEREE #3

First of all, we want to thank the reviewer for his useful comment and suggestion. For this reason, we have been incorporated it in the present revised version of the manuscript.

Researchers should expand their knowledge Alternatively, there are scholars' references on determining the ratio of samples to variables for SEM analysis.

ANSWER: We agree with the referee suggestion. We have uploaded the text as follows:

“To determinate the ratio sample to variables for SEM analysis is also important. We can find interesting literature in this field [106, 107, 108].”

References

[106] Wolf, E.J.; Harrintong, K.L.; Clark, S.L.; Miller, M.W. Sample size requirements for Structural Equation Model: An Evaluation of Power, Bias and Solution Propiety. Educ. Psycol. Meas. 2013, 73, 913-934.

[107] MacCallum, R.C.; Widaman, K.F.; Zhang, S.; Hong, S. Sample size in factor analysis. Psycol. Methods 1999, 4, 84-89.

[108] Gagné, P.; Hancock, G.R.; Measurement model quality, sample size, and solution propriety in confirmatory factor models. Multivariate Behav. Res. 2006, 41, 65-83.
